# Coherent structural trapping through wave packet dispersion during photoinduced spin state switching

Henrik T. Lemke[1,2], Kasper S. Kjær[3,4,5], Robert Hartsock[1,3], Tim B. van Driel[1,4], Matthieu Chollet[1], James M. Glownia[1], Sanghoon Song[1], Diling Zhu[1], Elisabetta Pace[6], Samir F. Matar[7], Martin M. Nielsen[4], Maurizio Benfatto[6], Kelly J. Gaffney[8], Eric Collet[9] & Marco Cammarata[9]

The description of ultrafast nonadiabatic chemical dynamics during molecular photo-transformations remains challenging because electronic and nuclear configurations impact each other and cannot be treated independently. Here we gain experimental insights, beyond the Born–Oppenheimer approximation, into the light-induced spin-state trapping dynamics of the prototypical $[Fe(bpy)_3]^{2+}$ compound by time-resolved X-ray absorption spectroscopy at sub-30-femtosecond resolution and high signal-to-noise ratio. The electronic decay from the initial optically excited electronic state towards the high spin state is distinguished from the structural trapping dynamics, which launches a coherent oscillating wave packet (265 fs period), clearly identified as molecular breathing. Throughout the structural trapping, the dispersion of the wave packet along the reaction coordinate reveals details of intramolecular vibronic coupling before a slower vibrational energy dissipation to the solution environment. These findings illustrate how modern time-resolved X-ray absorption spectroscopy can provide key information to unravel dynamic details of photo-functional molecules.

[1] Linac Coherent Light Source, SLAC National Accelerator Laboratory, Menlo Park, California 94025, USA. [2] SwissFEL, Paul Scherrer Institut, Villigen PSI 5232, Switzerland. [3] PULSE Institute, SLAC National Accelerator Laboratory, Stanford University, Menlo Park, California 94025, USA. [4] Molecular Movies, Department of Physics, Technical University of Denmark, Lyngby DK-2800, Denmark. [5] Department of Chemical Physics, Lund University, Box 124, Lund SE-22100, Sweden. [6] Laboratori Nazionali di Frascati-INFN, P.O. Box 13, Frascati I-00044, Italy. [7] ICMCB, CNRS UPR 9048, Univ. Bordeaux, Pessac F-33608, France. [8] SSRL and PULSE Institute, SLAC National Accelerator Laboratory, Menlo Park, California 94025, USA. [9] Univ. Rennes 1, CNRS, UBL, Institut de Physique de Rennes (IPR) - UMR 6251, F-35042 Rennes, France. Correspondence and requests for materials should be addressed to H.T.L. (email: henrik.lemke@psi.ch) or to M. Cammarata (email: marco.cammarata@univ-rennes1.fr).

D uring a photo-transformation in a molecule, an initial electronic excitation upon photon absorption launches a wave packet that dissipates energy into different electronic and vibrational degrees of freedom[1]. Thus such a wave packet can get trapped by structural reorganization in a new electronic state exhibiting modified properties, often in $<1$ ps. On this short timescale, the Born–Oppenheimer approximation separating wave functions of atomic nuclei and electrons is often invalid, undermining the precision of theoretical predictions and raising the importance of ultrafast investigations. This is well illustrated by the recent debate on the light-induced excited spin-state trapping (LIESST) process[2] intensively studied in the prototypical $[Fe(bpy)_3]^{2+}$ system[3–9]. $[Fe(bpy)_3]^{2+}$ (Fig. 1a, where bpy = 2,2′-bipyridine) forms a $^1A_{1g}$ low spin ground state (LS, $S = 0$), where the average bond length between Fe and the six N atoms of the bpy ligand is $r \sim 2.0$ Å (refs 3–6). Photoexcitation creates an electronic metal-to-ligand charge-transfer (MLCT) state, which rapidly decays to a metastable $^5T_{2g}$ high spin state (HS, $S = 2$) with unity quantum yield[10]. The population of antibonding $e_g$ orbitals in the HS state changes the Fe-N equilibrium distances to $r \sim 2.2$ Å (ref. 6), as also observed during LIESST in spin-crossover materials[11]. While it is clear that the HS state is reached on the 100 fs timescale and accompanied by coherent vibrations, the exact timescale, sequence of intermediates and the structural nature of such vibration are subjects of intense theoretical discussions[12,13]. A recent X-ray emission spectroscopic (XES) study (sensitive primarily to the electronic configuration) has evidenced a short passage via a $^3T$ state[5] although this picture has been debated based on an optical spectroscopic study[8]. In contrast to those experiments, lacking a direct structural probe, the femtosecond X-ray absorption near edge structure (XANES, see Methods section and Fig. 1b) is sensitive to local electronic and structural dynamics around the absorbing element (here Fe K-edge) and therefore well suited to track the processes at the origin of LIESST. The developments of bright X-ray sources[14,15] facilitate ultrafast XANES measurements at femtosecond time resolution and signal-to-noise unachievable until recently.

Here we show experimental data obtained at the Linac Coherent Light source from an $[Fe(bpy)_3]^{2+}$ molecular ensemble in aqueous solution (see Methods section). The observed dynamics clearly reveals the excitation into the MLCT state, which shows a 120 fs lifetime, followed by a coherent structural wave packet in the HS state whose nature can be assigned to molecular breathing. The dispersion dynamics of this wave packet is encoded in the transient XANES spectra and provides experimental details of the LIESST mechanism: The vibrational energy couples also to other intramolecular vibrations than breathing and stabilizes the spin transition. These incoherent processes can be distinguished from the dissipation of this vibrational energy to the environment.

## Results

### General considerations.
The recorded XANES spectra (Fig. 1c) of the $[Fe(bpy)_3]^{2+}$ LS ground state as well as at a time delay $t = 10$ ps after photoexcitation, when the HS state is already stabilized, agree with previous results at 100 ps time resolution[6]. The femtosecond transient XANES changes $\Delta I(t)/I_{off}$ after photoexcitation were measured at selected photon energies, ranging from the pre-edge region (7,113.5 eV), with reduced absorption in the HS compared to the LS state as $e_g$ states become occupied[16], to 7,164 eV, predominantly sensitive to molecular structure (minimal electronic contributions, Fig. 2a). Those traces, at or above the edge, clearly show damped oscillations for $t \gtrsim 200$ fs before converging within 3–6 ps (Fig. 1d and inset of

Fig. 2b) towards the HS/LS difference previously reported[6]. Time traces at 7,121.5 and 7,132.5 eV change sign at around 100 fs, clearly evidencing the presence of a short-lived intermediate.

Considering the similar features in the XANES transients at different energies, we analyse these globally (Fig. 2a) with a phenomenological model (Supplementary Note 1). It includes an initially photoinduced state (MLCT), which stochastically populates the final HS state where the system undergoes a damped coherent oscillatory motion (Fig. 2b and Supplementary Fig. 1). This simple model is able to reproduce all main features of the experimental data. The obtained initial XANES amplitudes of the intermediate match the expected formal $Fe^{3+}$ MLCT spectrum (Fig. 1d) approximated by shifting the LS spectrum by 1 eV, $I_{MLCT}/I_{LS} = I_{LS}(E + 1 \text{ eV})/I_{LS}(E)$, as suggested by previous XANES studies on similar compounds[17,18]. The lifetime $\tau_{MLCT} = 120(10)$ fs of this clear fingerprint of the MLCT intermediate is compatible with the lifetime found by XES[5].

The observed 126(3) $cm^{-1}$ vibration frequency in the HS state has also been found to modulate transient optical absorption data and has been assigned to a non-totally symmetric Fe-N stretching and bending mode[7,8]. As the observed vibration modulates XANES, mainly sensitive to the average first-neighbour distances (Fe-N), it must be directly associated with the reaction coordinate of LIESST and identifying its nature is therefore crucial for its understanding.

### Symmetry considerations of the vibrational modes.
Molecular vibration frequencies calculations were carried out for $[Fe(bpy)_3]^{2+}$ in the HS state, after geometry optimization (see Methods section). Figure 3 shows the vibrational modes of specific interest for interpretation of the observed XANES oscillations. For the $FeN_6$ system of $O_h$ point symmetry, there are six stretching modes: two $e_g$, three $t_{1u}$ and one totally symmetric breathing mode ($a_{1g}$). Rigorously speaking, the symmetry of $[Fe(bpy)_3]^{2+}$ is not $O_h$ but $D_3$ and the Fe-N vibrations couple with ligand modes. Therefore, the description of the modes is more difficult. However, we can distinguish main characters of the modes in terms of Fe-N elongation or ligand torsion and describe them in the almost $O_h$ symmetry for simplicity[12]:

The $a_{1g}$ breathing mode calculated at 124.4 $cm^{-1}$ is associated with in-phase stretching of the 6 Fe-N bonds with almost rigid ligands (Fig. 3a,b and visualized in Supplementary Movie 1). A similar $a_{1g}$ stretching mode was calculated by density functional theory at 121.4 $cm^{-1}$ by Sousa et al.[12]. The totally symmetric elongation of the 6 Fe-N bonds from 2.0 to 2.2 Å is also the main reaction coordinate in terms of structural change between LS and HS states. With this mode, the six Fe-$N_i$ bond lengths $r_i$ oscillate in phase by $x_i$ around 2.2 Å (breathing mode $q_B$, Fig. 3a) with $r_i = x_i + 2.2$ Å. Since this mode transforms similar to $q_B = x_1 + x_2 + x_3 + x_4 + x_5 + x_6$, the reaction coordinate $r = <r_i>$ oscillates as illustrated in Fig. 3g. We identify another $a_{1g}$ mode at 353.4 $cm^{-1}$ with important bpy stretching (Fig. 3f), mainly through the C=C bond, coupled with Fe-N stretching. This mode is too high in frequency to be associated with the oscillations observed in XANES. However, it may contribute to the energy dissipation and broadening of the Fe-N distribution as discussed later in more detail.

We find other modes close to the 126(3) $cm^{-1}$ frequency of the oscillation observed in XANES. Modes at 114.8 and 115.9 $cm^{-1}$ have $e_g$ symmetry and Fe-N stretching character (Fig. 3c,d) and are similar to the modes found at 116.5 and 116.2 $cm^{-1}$ by Sousa et al.[12]. The stretching mode at 114.8 $cm^{-1}$

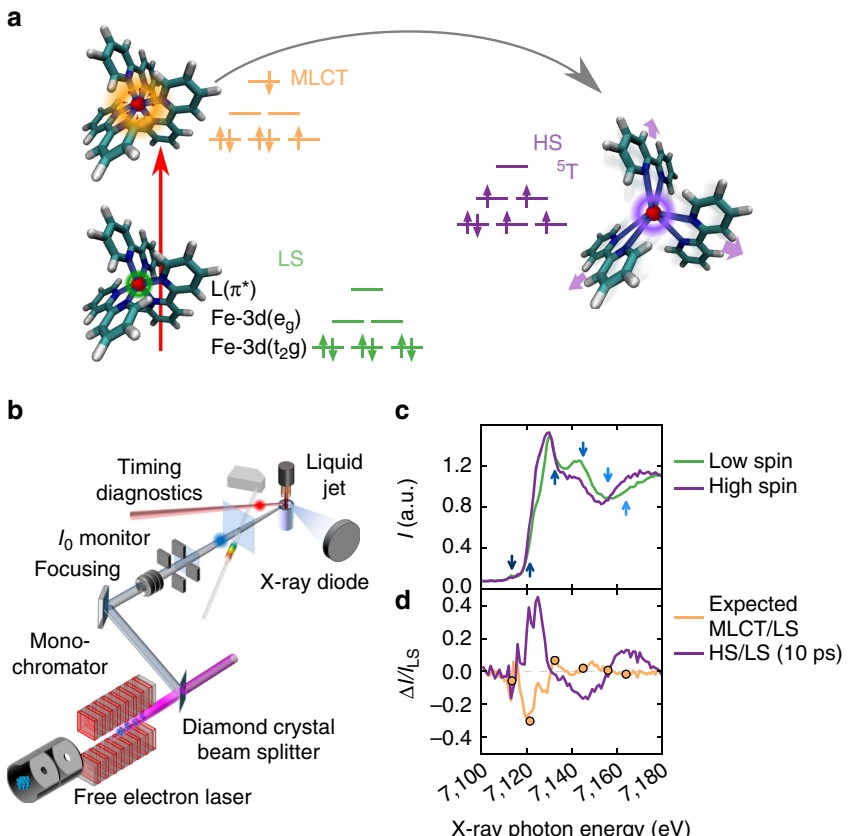

**Figure 1 | X-ray absorption fingerprints of molecular transformation.** (**a**) Schematic light-induced excited spin-state trapping for $[Fe(bpy)_3]^{2+}$, where the Fe (red) is bonded to six N (blue) of the bpy ligands (L). After the initial photoexcitation of the LS ($t_{2g}^6 e_g^0 L^0$) state into an MLCT state ($t_{2g}^5 e_g^0 L^1$), the system decays towards the HS ($t_{2g}^4 e_g^2 L^0$) state. (**b**) Schematic experimental setup with 25(5) fs RMS time resolution. The X-ray beam from a free electron laser is monochromatized by a double diamond (111) crystal and focused to $\sim 10\,\mu m$ by means of Beryllium X-ray lenses for probing $[Fe(bpy)_3]^{2+}$ dissolved in water. (**c**) Changes between the LS and HS XANES spectra. Arrows indicate photon energies for which high time resolution data have been measured. (**d**) The HS/LS spectra change ratio (magenta line, measured at 10 ps) and the expected ratio between MLCT and LS state (orange solid line) calculated as a $+1\,eV$ shift of the MLCT spectrum with respect to the measured LS one. The dots in (**d**) are the measured MLCT amplitude based on a global fit of the data (see text and Fig. 2). Both curves have been scaled to 100% conversion.

transforming similar to $q_{s1} = -x_1 + x_2 - x_4 + x_5$ is associated with out-of-phase Fe-N stretching along $x$ and $y$, keeping so $r$ constant in time (Fig. 3h). The mode at $115.9\,cm^{-1}$ transforms similar to $q_{s2} = -x_1 - x_2 + 2x_3 - x_4 - r_5 + 2x_6$ and is associated with in-phase Fe-N stretching along $x$ and $y$ and twice larger out-of-phase stretching along $z$, also keeping $r = 2.2\,\text{Å}$ constant in time (Fig. 3i). Therefore, these modes cannot explain the observed XANES oscillations, especially at energies where the XANES signal changes linearly with $r$ (see following section). Another mode found at $138.8\,cm^{-1}$ ($t_{1u}$ symmetry, calculated by Sousa *et al.*[12] at $132.7\,cm^{-1}$) corresponds to ligand bending with no Fe-N elongation and it cannot explain XANES oscillations mainly sensitive to Fe-N distance. Further calculated modes differ too much in frequencies ($< 100\,cm^{-1}$ or $> 150\,cm^{-1}$) to be attributed to the oscillations observed in XANES.

The contribution of the LS breathing mode, coherently activated by an impulsive Raman process, to the XANES oscillation can also be questioned. This possibility is excluded because on the one hand, the LS breathing frequency is significantly higher ($145\,cm^{-1}$, from Sousa *et al.*[12]) and on the other hand, an impulsive process would cause sine-like oscillations around $r = 2.0\,\text{Å}$ and around $\Delta I/I = 0$ in XANES, which is not the case here. Moreover, modulations at the same frequency have been observed in optical spectroscopy around the

excited state absorption but not in the ground state bleaching part of the spectrum[7].

The characteristic oscillation frequency observed here by XANES or previously by optical spectroscopy can now be firmly assigned to the breathing mode in the HS state, thereby identifying the nature of the nuclear reaction coordinate. This highlights the importance of a direct and local structural probe such as XANES to understand the nature of coherent structural dynamics.

**Multiple scattering calculations.** More details of the process were obtained through XANES multiple scattering calculations of the molecular structure and varying $r$ from 1.7 to 2.5 Å, together with different degrees of electronic contributions (chemical shift, Supplementary Note 2). At $E = 7,145\,eV$, the signal is roughly linear with $r$ and relatively independent of the chemical shift (Fig. 4a and Supplementary Fig. 2). We can thus interpret this time-resolved signal as proportional to the average change of $r$. The amplitude of the oscillation is significantly smaller than the $\sim 0.2$ Å bond elongation from LS to HS, which indicates that much of the coherence of the initially excited wave packet is already lost 200–300 fs after photoexcitation. We attribute this effect of decoherence to wave packet spreading during MLCT-to-HS conversion. The $\sim 120$ fs lifetime of the MLCT intermediate reduces the ensemble average amplitude of the oscillations

significantly (as shown in Fig. 2b) and causes an apparent ∼50 fs phase shift. The latter was also observed in the recent optical spectroscopy work and has been attributed to a sub-50-fs time

constant for the MLCT-to-HS transition[8]. Such fast transition, however, is also incompatible with the purely structural information in our data (see 'Transient molecular distribution model' section).

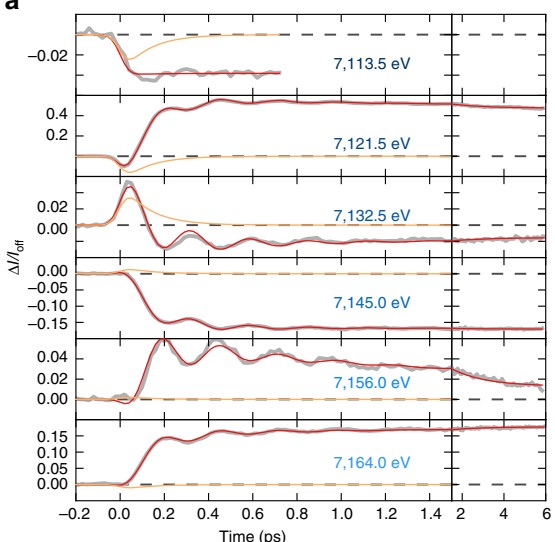

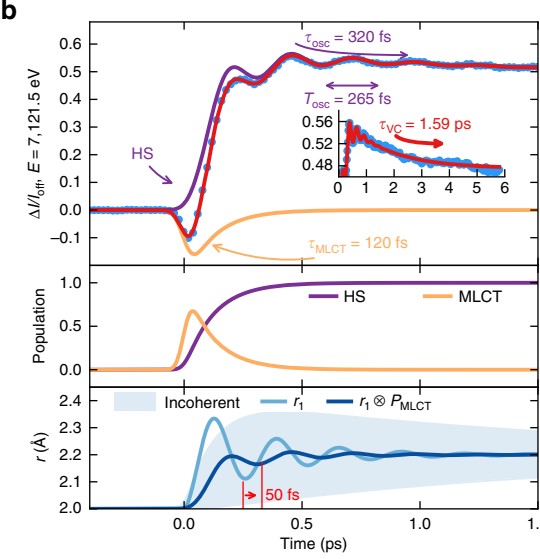

**Figure 2 | Following dynamics in real time.** (**a**) Time scans of relative absorption change at selected X-ray energies (solid grey lines) revealing 126(3) cm$^{-1}$ oscillations (265 fs period). Red lines correspond to the global fit of the entire data set with an empirical model (Supplementary Information). While all amplitude parameters have been varied for every X-ray energy, the dynamic parameters are the same for all energies: MLCT lifetime $\tau_{MLCT}=120(10)$ fs, oscillation period $T_{OSC}=265(10)$ fs and damping $\tau_{OSC}=320(10)$ fs and vibrational cooling $\tau_{VC}=1.6(0.1)$ ps. Orange lines represent the MLCT contributions. (**b**) Example fit for 7,121.5 eV (top panel), showing the individual contributions of MLCT and oscillating HS contributions along with the total signal. The inset shows the data on a longer time window. The model disentangles the electronic kinetic description (MLCT and HS population, mid panel) from the structural dynamics given by the time evolution of the Fe-N distance $r$ (bottom panel). The exponential growth of the HS population from the MLCT intermediate state leads to an average coherent oscillating trajectory $\bar{r}(t)=r_1(t)\otimes P_{MLCT}(t)$. This has a reduced amplitude and apparent phase shift (∼50 fs) with respect to a directly initiated damped oscillating trajectory $r_1(t)$ in the HS potential. The incoherent part of the molecular oscillations, the transient distribution width in $r$, decays within 1.6 ps.

**Wave packet delocalization.** Additionally to the average structure, the degree of delocalization along $r$ in the HS potential, described by the ensemble distribution width, can be monitored nearly independently in signals that change markedly nonlinearly with $r$ (similar to 7,156 eV, Fig. 4a and Supplementary Fig. 2). The difference signal at 7,156 eV (close to HS/LS isosbestic point) can be expressed to first order as function of the root mean square (RMS) width ($\sigma_r$) of the distribution in $r$ (Fig. 4b and Supplementary Fig. 3). We approximate the extended X-ray absorption fine structure (EXAFS) equation as resulting only from the first coordination shell with six N atoms at the same distance $r$: $\chi(k,r)=\frac{6f(k)}{kr^2}\exp(-2k^2\sigma_r^2)\sin(2kr)$, where $f$ and $k$ denote the scattering amplitude and electron wavenumber, respectively, broadened by a Gaussian function. After expansion around the interference minimum in the HS state, one obtains (Supplementary Note 3): $\Delta I/I \simeq -44.5\sigma_r^3$, shown as right scale of Fig. 4b.

The initial fast increase in the 7,156 eV trace, where equilibrium HS and LS states have same XANES signals (Fig. 1c), therefore supports the strong dispersion in $r$ induced during the MLCT-to-HS transition. This dispersion in $r$ can also be related to the activation of other stretching vibrations, similar to the $e_g$ modes mentioned above, which contrary to the breathing mode keep $r$ constant but distribute the lengths of the six individual bond Fe-N$_i$ around $r\sim2.2$ Å (Fig. 3).

The relaxation mechanisms of the coherent molecular breathing are characterized by a damping constant $\tau_{OSC}=320(10)$ fs of the oscillation, accompanied by a slower $\tau_{VC}=1.6(0.1)$ ps decaying component. As $\tau_{OSC}$ is not related to a significant decrease in $\sigma_r$, it can be associated with decoherence of the wave packet without significant loss of vibrational energy with respect to coordinate $r$. In contrast, $\tau_{VC}$ directly reflects narrowing of the ensemble distribution around $r\sim2.2$ Å, as the system relaxes in the HS potential and passes vibrational energy onto the environment. The timescale agrees with solvent temperature and density rise observed by time-resolved scattering[19]. This constitutes a direct observation of the vibrational cooling process, selectively probing the molecular breathing reaction coordinate $r$.

**Transient molecular distribution model.** The phenomenological fit and XANES calculation provide direct semi-quantitative insight into the structural dynamics of the molecule, suggesting in particular that once in the HS state, the wave packet dispersion along $r$ happens significantly earlier than the vibrational cooling related to energy dissipation. In order to test the consistency of the separate findings, we approximate the propagating wave packet by a classical model where stochastically generated 'inelastic events' change the phase and the energy of an ensemble of molecular trajectories (Supplementary Note 4), thereby obtaining a time-dependent number density in $r$: $g(r,t)$ (Fig. 5c). The energy transfer describes energy redistribution both to intramolecular vibrations and the solvent as suggested by Veenendaal et al.[13]. Using the multiple scattering calculations discussed above, we simulate the expected time-dependent XANES signals from this distribution as:

$$I(t,E)=\int S(E,r)g(r,t)\mathrm{d}r \qquad (1)$$

where $S(E,r)$ are the absorption spectra obtained from the multiple scattering calculations mentioned above. The time variation $\Delta I / I_{off}$ at the observed photon energies is shown in red in Fig. 4b and Supplementary Fig. 3. The qualitatively superior agreement of a calculation taking into account

this temporal distribution $g(r,t)$ over a calculation using its time-dependent average $\bar{r}(t)$ substantiates XANES' sensitivity to the structural distribution along the reaction coordinate $r$.

The expected structural XANES contribution of the 70 fs $^3T$ intermediate observed by XES[5] is dominated by changes along $r$ (ref. 12) and therefore cannot be distinguished from the HS contribution as independent fingerprint in the XANES spectrum. Its intermediate population, however, has an influence on the structural distribution and was therefore tested by a more elaborate analysis, adding a Triplet state with a lifetime of 70 fs to the transient molecular distribution model. This has been achieved in a similar manner as described for the MLCT state (that is, by a transition time of 70 fs, exponentially distributed). The resulting distributions, average positions and populations are shown in Fig. 5 along with the results for the direct MLCT → 120 fs → HS process. The experimental data shown in the figure corresponds to the data at 7,145 eV rescaled to match the observed 0.2 Å change. The main difference between the two calculations is the smaller oscillation amplitude when the Triplet is added. This is expected since it acts as another source of dephasing. While this result is not an independent additional proof of the population of the $^3T$ state, it suggests that the $^3T$ contributes to the decoherence induced by the MLCT-HS transition.

Using the distribution model, the structural influence of the proposed sub-50-fs MLCT state lifetime[8] can be simulated and be compared to the experimental data. The results clearly show that the expected amplitude of oscillations would be significantly higher than measured, through a higher initial coherence of the wave packet (Supplementary Fig. 4).

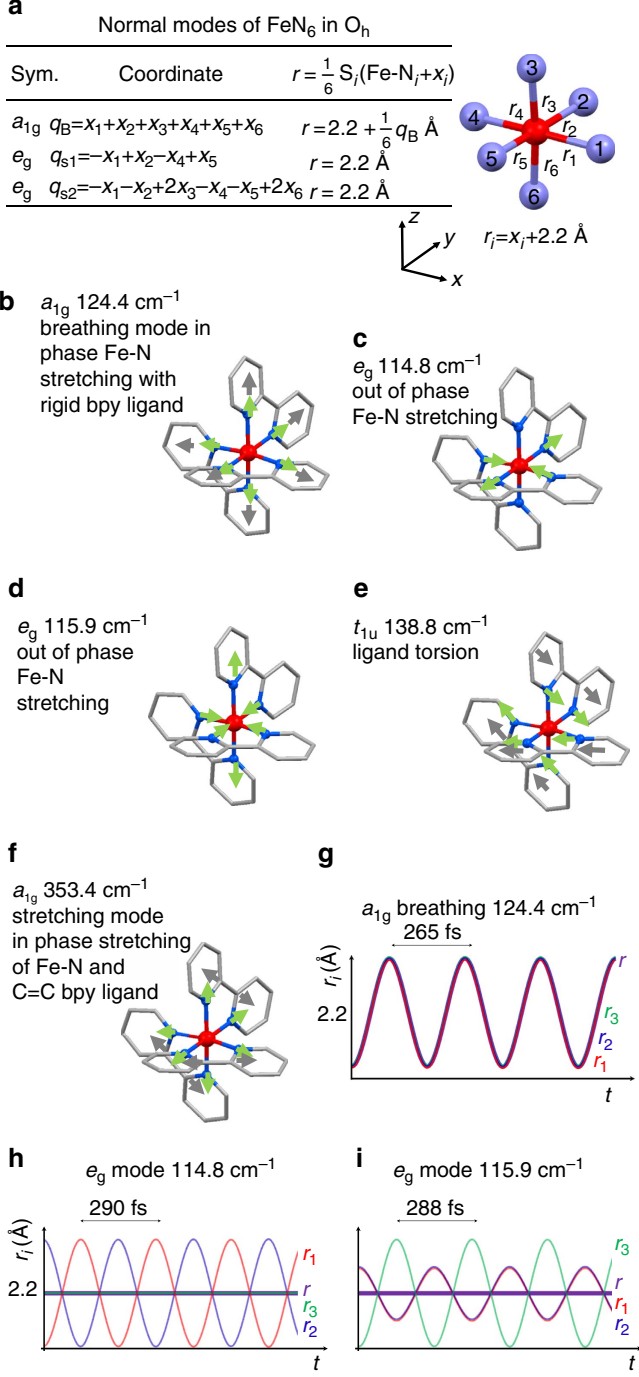

**Figure 3 | HS normal modes.** (**a**) Fe-N bond coordinate and average length $r$ for $a_{1g}$ and $e_g$ symmetry normal modes. (**b–f**) Respective representation of normal breathing mode (124.4 cm$^{-1}$, in-phase stretching of the 6 Fe-N$_i$ bonds with almost rigid bpy ligands), the global stretching mode (353.4 cm$^{-1}$, bpy and Fe-N stretching), stretching modes at 114.8 and 115.9 cm$^{-1}$ and a bending mode at 138.8 cm$^{-1}$. Green arrows show N motions and grey ones ligand motions. (**g–i**) Time evolution of the symmetry independent bonds $r_1$, $r_2$ and $r_3$ with respect to inversion symmetry and their average $r$ (purple).

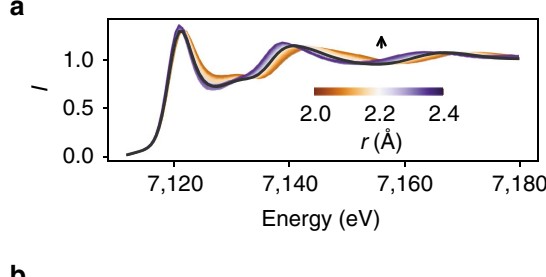

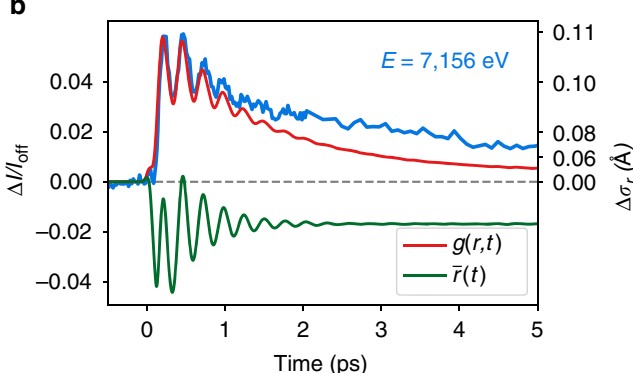

**Figure 4 | Coherent versus incoherent structural dynamics.** (**a**) XANES spectra calculated for a variety of Fe-N distances $r$ from 2 Å (blue) to 2.4 Å (gold). (**b**) experimental data at 7,156 eV (blue), together with calculated signals for the distance distribution $g(r,t)$ (red) of an ensemble motion model (red in Fig. 5c) or its average value $\bar{r}(t)$ (green). The data are significantly better reproduced by simulations, including the evolving ensemble width distribution. The right $y$ scale represents the calculated width change $\Delta\sigma_r$ of a normal distributed approximation of $g(r,t)$ based on the expansion of the EXAFS equation.

Such short MLCT lifetime is therefore incompatible with our structural observations at least in the framework of our interpretation.

## Discussion

In summary, our transient data enable to draw a rather complete picture of ultrafast LIESST (Fig. 6) by taking advantage of the varying sensitivity to electronic, coherent and incoherent structural

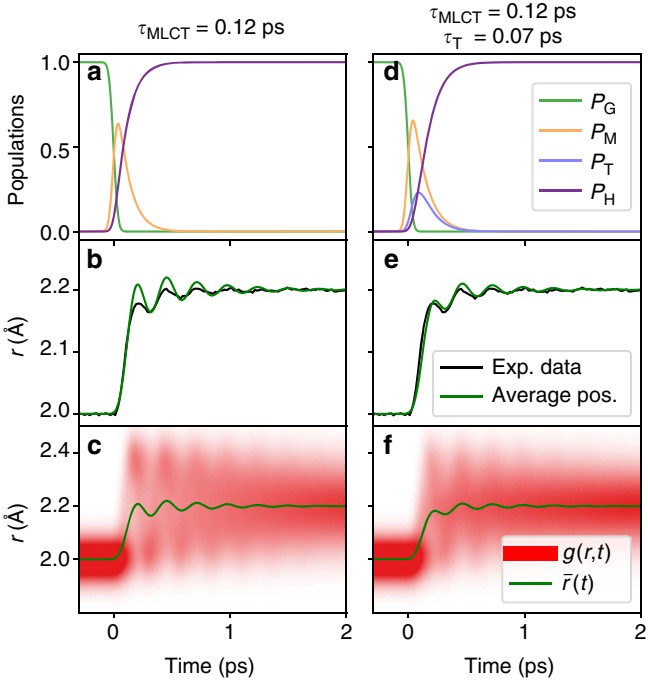

**Figure 5 | Contribution of intermediates to transient molecular distribution.** Simulations of the transient molecular distribution in coordinate $r$ for MLCT→HS ($\tau_{MLCT} = 120$ fs) and MLCT→3T→HS ($\tau_T = 70$ fs) electronic state transition (**a–c,d–f**, respectively). The time-dependent populations of the different electronic states are shown in panels (**a** and **d**), which give rise to the calculated distribution $g(r, t)$ and average coordinate $r$: $\bar{r}(t)$ (**c,f**). The experimental data at 7,145 eV (blue) scaled in $r$ has additionally been overlaid with the simulated signal (**b,e**).

changes over the XANES spectrum, measured at unprecedented time resolution and signal/noise. Our results and interpretation are consistent with the reported data, link individual electronic with structural observations from femtosecond to picosecond timescale and extend the understanding of LIESST by the key role of activation and damping of structural vibrations in the trapping process (Fig. 6a). We establish a MLCT-to-HS conversion as electron–phonon coupling process at 120 fs time constant by observation of electronic and structural information. The result is compatible with reports including a short-lived $^3$T intermediate based on electronic state sensitivity[5]. The structural response upon population of antibonding $e_g$ orbitals, shifting the minimum of the potential energy curve towards longer Fe-N bonds, is accompanied by the directly measured strong dispersion of the excited wave packet and activates a damped oscillatory signal of structural origin. Our local probe allows to solve the ambiguity in assigning this repeatedly measured coherent molecular vibration[7,8,20] by cross checking with density functional theory simulations and symmetry arguments. Thereby, the main reaction coordinate, generally introduced by Hauser and colleagues[2] to describe photoinduced spin-state switching, is identified here as the molecular breathing of the Fe-N$_6$ octahedron with rigid bpy ligands. The directly probed fast decoherence in the transient wave packet (Fig. 6b) at remaining wide distribution in $r$ underlines the strong coupling to a multitude of intramolecular vibrational modes that affect the Fe-N bonds. For example, the activations of low symmetry modes, similar to $e_g$ stretching ones at 114.8 cm$^{-1}$ and 115.9 cm$^{-1}$, do not contribute to the time dependence of $r$ but contribute to its broadening through the distribution of the Fe-N$_i$ bond lengths (Fig. 3). This process has been proposed by Veenendaal et al.[13] as prerequisite for the observed high quantum efficiency of the LIESST process. The fast damping of molecular breathing by energy dissipation towards other modes precludes recurrence towards the initial MLCT and thereby limits the MLCT-to-HS transition timescale to a half-cycle of the breathing mode, which is in good agreement with our observations.

The delayed narrowing of this distribution at longer timescale (1.6 ps) can be related to intermolecular coupling to solvent molecules that vibrationally cool the excited [Fe(bpy)$_3$]$^{2+}$ in the HS state. The good agreement of the time constant observed here for the vibrational cooling with the rise time of solvent temperature and density observed by X-ray scattering (1.1(0.3) ps)[19], suggest a rather direct coupling to the solvent.

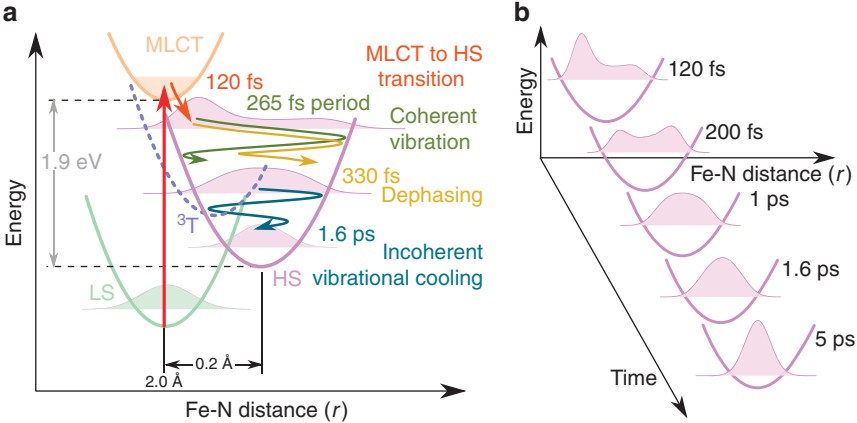

**Figure 6 | Structural trapping dynamics.** (**a**) Schematic representation of the structural trapping during the light-induced spin-state conversion in [Fe(bpy)$_3$]$^{2+}$ from LS to HS state along the Fe-N distance reaction coordinate $r$. The photoexcited MLCT (manifold) decays, through the $^3$T state ($t_{2g}^5 e_g^1 L^0$), towards the HS state within (120(10) fs) and a large fraction of energy is dissipated. It expands and coherently oscillates (breathing mode, 265 fs period) around the HS equilibrium structure while losing energy. The wave packet disperses at 330 fs time constant and vibrationally cools inside the HS state potential within 1.6 ps. (**b**) Schematic time evolution of the wave packet in the HS potential based on the simulated distribution model.

More generally, this prototypical study demonstrates the capabilities of state-of-the-art femtosecond XANES to bring the understanding of photoinduced chemical dynamics beyond simple 'molecular movies'. In addition to providing real-space movies of molecular motion on sub-Ångstrom and femtosecond scales during a photo-transformation[21], we show here that is now possible to reveal coherent and incoherent ultrafast dynamics along with the change of electronic state and what additional insight is gained from the ultrafast dispersion of the excited wave packet. The complexity of the molecular environment can be successfully reduced by a site-specific probe as XANES. Such real-time measurements of ultrafast energy redistribution into electronic and structural degrees of freedom can provide key information to understand a multitude of chemical[22], physical[23,24] and biological light-induced phenomena[25].

## Methods

**Experimental.** The time resolved X-ray absorption signal of aqueous solution of $[Fe(bpy)_3]^{2+}$ was measured using the pump/probe technique through total fluorescence detection[4]. We used a C*(111) double crystal monochromator available at the X-ray Pump Probe station, Linac Coherent Light source[26]. The $[Fe(bpy)_3]^{2+}$ complex was dissolved in water (concentration 30 mM) and circulated via a closed loop system through a 30 μm liquid jet. Such thin sample minimized the effect of temporal broadening due to group velocity mismatch between optical and X-ray beams ($\sim 1\,fs\,\mu m^{-1}$). The sample was excited by 530 nm pulses from a Ti:sapphire laser system and an Optical Parametric Amplifier (OPerA Solo, Coherent). The relative X-ray to optical pulse arrival time was recorded using the timing tool diagnostic[15]. The excitation fluence was set below the onset of nonlinear signal response at 2.2 mJ mm$^{-2}$. The overall time resolution was found to be $\sim 25\,fs$ RMS (that is, 60 fs full-width at half-maximum) by the phenomenological fit. It is close to the expected value when considering the convolution of pump laser pulse length ($\sim 50\,fs$), the X-ray pulse ($\sim 20\,fs$), the jitter correction precision ($\sim 10\,fs$) and $\sim 30\,fs$ due to group velocity mismatch between pump and probe pulses in the 30 μm sample. From the difference signal in Fig. 1d, we estimate that 75% of the molecules are excited into the MLCT state. The RMS XANES signal errors, determined statistically from repetitive FEL pulses inside the 20 fs rebinning interval, range between 0.06 and 0.15% of the signal levels.

**Theory.** XAS calculations used the distribution width in the LS state found by Daku and Hauser[9] and were performed using the MXAN code[27,28]. More details are given in Supplementary Note 2. Molecular vibration frequencies calculations were carried out for $[Fe(bpy)_3]^{2+}$ in the HS state by using hybrid B3LYP functional with LANL2DZ ECP basis set within Gaussian09 code[29]. Frequencies are determined from the second derivatives of the energy with respect to the atomic positions and then operating transformation to mass-weighted coordinates. Exploring the results especially for the vibrations and their animations with screen captures was done with Gaussview05 annex module to Gaussian09.

**Data availability.** All relevant data and programs are available from the authors upon request.

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

## Acknowledgements

Portions of this research were carried out at the Linac Coherent Light Source (LCLS) at the SLAC National Accelerator Laboratory. LCLS is an Office of Science User Facility operated for the U.S. Department of Energy Office of Science by Stanford University. M. Cammarata and E.C. thank ANR (ANR-13-BS04-0002) and Centre National de la Recherche Scientifique (CNRS) (PEPS SASLELX) for financial support. K.S.K., T.b.v.D. and M.M.N. acknowledge support from DANSCATT. K.S.K. gratefully acknowledges support from the Carlsberg Foundation. K.J.G. acknowledges support from the AMOS program within the Chemical Sciences, Geosciences and Biosciences Division of the Office of Basic Energy Sciences, Office of Science, U.S. Department of Energy. We thank Dan De Ponte (LCLS) for help with the sample delivery system. M.M.N. acknowledges support from The Danish Council for Independent Research under grant DFF – 4002-00272.

## Author contributions

H.T.L. and M. Cammarata conceived the project and analysed the data. H.T.L., K.S.K, K.J.G., E.C. and M. Cammarata set the physical picture for interpreting the data. H.T.L, K.S.K., R.H., T.b.v.D., M. Chollet, J.M.G., S.S., D.Z., E.C. and M. Cammarata performed the femtosecond XAS experiment. E.P. and M.B. performed the multiple scattering calculations. S.F.M. performed density functional theory calculations. M. Cammarata developed the molecular distribution model and simulated the transient data. H.T.L., E.C. and M. Cammarata wrote the paper. All authors contributed to discussions and gave comments on the manuscript.

**Additional information**

**Competing interests:** The authors declare no competing financial interests.

