## [Peer Review File · Nature Communications]

Reviewers' comments:

Reviewer #1 (Remarks to the Author):

As I mentioned in my previous review, this is a special team of researchers that are working at the cutting edge of a very important field at the interface of chemistry and physics. The present contribution is a significant improvement over the original version of this manuscript that I saw previously insofar as it is more clearly written, more focused, and more effectively highlights the new information gleaned from the (wonderful) experiments that have been performed. The quality of the data is remarkable; these measurements are really a tour de force for extremely challenging experimental studies. In addition, the computational work really helps to underscore several of the points the authors are extracting from the data analysis. I have to say that I was surprised - but am now convinced - of the mode(s) that are likely contributing to the observed dynamics in a way that I wasn't before. I learned something from this new version. I'm sure those results were there previously, but now they stand out. So, my compliments to the authors.

I do not wish to stand in the way of this being published (which I already have to some extent due to the delay in returning these comments - apologies for that), so I will just raise two points:

(1) I really do feel that the authors are shortchanging their work by trying to fit this into the format of this journal. There is a wealth of valuable information in the SI that simply will not be examined or appreciated by the majority of people who read this paper. It's obviously the authors' call, but I would have opted to fold most of that into a full paper and sent it to JACS or Chemical Science rather than forcing this into a communication format. I believe that paper would have far greater resonance in the community that this one will. Again, that's obviously the authors' (and editor's) call.

(2) This may be a consequence of the requirements of the journal (I don't know as I have never published in a Nature journal before), but the referencing is rather strange. The one that stood out for me was using references 12 and 13 as the touchstones for work on the ultrafast dynamics of Fe(II) polypyridyl complexes. I mean no disrespect to these papers by any means, but they are hardly representative of the benchmark papers in this field. There are other instances of this in the ref list as well. I'm not suggesting the authors provide comprehensive lists of references - this is not a review, after all - but when one is limited in terms of numbers, one needs to make judicious choices. Just a suggestion.

Overall, I would rather see a full paper that includes a more extended discussion of the analyses presented in the SI, but I leave this decision to the authors and the editor. I will simply sit back and admire their work!

Reviewer #2 (Remarks to the Author):

This is a report of ultrafast time-resolved XANES spectroscopy at sub-30-femtosecond resolution and high signal to noise ratio. By utilizing the spectroscopic technique, the authors focus on ultrafast transformations of a hexa-coordinated iron complex where both molecular structure and electronic state evolve significantly when moving from an optically excited state to a final photoproduct. In this case, the Born-Oppenheimer approximation is not valid as electronic and structural degrees of freedom interact strongly. Based on their experimental findings, the decay from the initial excited state towards the high spin state is clearly detected, which launches a coherent oscillating wave packet (265 fs period). The coherent vibration mode at 124.4 cm⁻¹ was assigned to totally symmetric breathing mode in the HS state based on their DFT calculations. This paper is reporting very significant technical advances and potentially worthy of publication in the Nature Communications.

My reservations are as follows.

1) Have the authors examined the possibility of assignment of 124.4 cm⁻¹ as the vibration mode of LS ground state? Since XANES detects both ground and excited states, such possibility need to be taken into account.

2) If the assignment of the coherent vibration mode by the authors (in-phase breathing of the rigid bipy ligands through simultaneous stretching of the six Fe-N bonds) is correct, the symmetry of the iron complex changes from Oh to D3 with the vibration. In this case, not only Fe-N bond lengths but also N-Fe-N angles are also modulated in time, which makes interpretation of the reaction coordinate more complicated (Figure 4a). How is this issue implemented in the discussions?

Response to Referees

In the following the Reviewer comments are answered point-to-point. Where applicable, changes in the manuscript text that are related to reviewer's comments have been marked for easier tracking in this letter and in the manuscript ([Rx.x]).

Reviewer 1

As I mentioned in my previous review, this is a special team of researchers that are working at the cutting edge of a very important field at the interface of chemistry and physics. The present contribution is a significant improvement over the original version of this manuscript that I saw previously insofar as it is more clearly written, more focused, and more effectively highlights the new information gleaned from the (wonderful) experiments that have been performed. The quality of the data is remarkable; these measurements are really a tour de force for extremely challenging experimental studies. In addition, the computational work really helps to underscore several of the points the authors are extracting from the data analysis. I have to say that I was surprised - but am now convinced - of the mode(s) that are likely contributing to the observed dynamics in a way that I wasn't before. I learned something from this new version. I'm sure those results were there previously, but now they stand out. So, my compliments to the authors.

I do not wish to stand in the way of this being published (which I already have to some extent due to the delay in returning these comments - apologies for that), so I will just raise two points:

We thank the reviewer for the nice comments about our paper. Indeed we have invested so much energy in this work that is a pleasure to know that the results are considered novel and very interesting.

(1) I really do feel that the authors are shortchanging their work by trying to fit this into the format of this journal. There is a wealth of valuable information in the SI that simply will not be examined or appreciated by the majority of people who read this paper. It's obviously the authors' call, but I would have opted to fold most of that into a full paper and sent it to JACS or Chemical Science rather than forcing this into a communication format. I believe that paper would have far greater resonance in the community that this one will. Again, that's obviously the authors' (and editor's) call.

Response: we thank the referee for the suggestion. As Nature Communication allows longer articles, we have followed your suggestion and moved a good fraction of the Supplementary materials (DFT calculation, symmetry of the mode, contribution of the triplet state) in the main text. We have kept derivation and technicalities in the supplementary materials as they might be less interesting for the casual reader. We have marked the changes as **R1.1**

(2) This may be a consequence of the requirements of the journal (I don't know as I have never published in a Nature journal before), but the referencing is rather strange. The one that stood out for me was using references 12 and 13 as the touchstones for work on the ultrafast dynamics of Fe(II) polypyridyl complexes. I mean no disrespect to these papers by any means, but they are hardly representative of the benchmark papers in this field. There are other instances of this in the ref list as well. I'm not suggesting the authors provide comprehensive lists of references - this is not a review, after all - but when one is limited in terms of numbers, one needs to make judicious choices. Just a suggestion.

Referee is right, but there are already many references in the text about Fe(II) polypyridyl complexes. The part of the text where ref 12 and 13 were mentioned was referring to theoretical models. We therefore rewrote the this part of the text "the structural nature of such vibration are subjects of intense theoretical discussions^{12,13}".

[Changes R1.2]

Overall, I would rather see a full paper that includes a more extended discussion of the analyses presented in the SI, but I leave this decision to the authors and the editor. I will simply sit back and admire their work!
We thank the reviewer again. We hope that she/he will appreciate the revised version.

Reviewer 2

This is a report of ultrafast time-resolved XANES spectroscopy at sub-30-femtosecond resolution and high signal to noise ratio. By utilizing the spectroscopic technique, the authors focus on ultrafast transformations of a hexa-coordinated iron complex where both molecular structure and electronic state evolve significantly when moving from an optically excited state to a final photoproduct. In this case, the Born-Oppenheimer approximation is not valid as electronic and structural degrees of freedom interact strongly. Based on their experimental findings, the decay from the initial excited state towards the high spin state is clearly detected, which launches a coherent oscillating wave packet (265 fs period). The coherent vibration mode at 124.4 cm⁻¹ was assigned to totally symmetric breathing mode in the HS state based on their DFT calculations. This paper is reporting very significant technical advances and potentially worthy of publication in the Nature Communications.

We thank the reviewer for the positive comment on our manuscript

My reservations are as follows.

1) Have the authors examined the possibility of assignment of 124.4 cm⁻¹ as the vibration mode of LS ground state? Since XANES detects both ground and excited states, such possibility need to be taken into account.

We have added a sentence in the manuscript (see point R2.1) that states that both DFT (a_{1g} mode at higher frequency for the LS state) and optical experiments (oscillation observed in the excited state absorption part of the spectrum) strongly support our assignment to the HS state

2) If the assignment of the coherent vibration mode by the authors (in-phase breathing of the rigid bpy ligands through simultaneous stretching of the six Fe-N bonds) is correct, the symmetry of the iron complex changes from Oh to D3 with the vibration. In this case, not only Fe-N bond lengths but also N-Fe-N angles are also modulated in time, which makes interpretation of the reaction coordinate more complicated (Figure 4a). How is this issue implemented in the discussions?

We have extended the discussion about the modes and stated (point R2.1) that despite the fact that (as the reviewer is correctly stating) the symmetry is D3, it is close enough to Oh that can be used to get a qualitative picture as shown in ref 12.

REVIEWERS' COMMENTS:

Reviewer #2 (Remarks to the Author):

I found my reservations are properly answered by the authors in the revised manuscript. I think the revised MS is now appropriate for publication in the Nature Communications.